# Do Microplastics Have Neurological Implications in Relation to Schizophrenia Zebrafish Models? A Brain Immunohistochemistry, Neurotoxicity Assessment, and Oxidative Stress Analysis

**DOI:** 10.3390/ijms25158331

**Published:** 2024-07-30

**Authors:** Alexandra Savuca, Alexandrina-Stefania Curpan, Luminita Diana Hritcu, Teodora Maria Buzenchi Proca, Ioana-Miruna Balmus, Petru Fabian Lungu, Roxana Jijie, Mircea Nicusor Nicoara, Alin Stelian Ciobica, Gheorghe Solcan, Carmen Solcan

**Affiliations:** 1Doctoral School of Geosciences, Faculty of Geography and Geology, “Alexandru Ioan Cuza” University of Iasi, Carol I Avenue, 20A, 700505 Iași, Romania; alexandra.savuca@yahoo.com; 2Doctoral School of Biology, Faculty of Biology, “Alexandru Ioan Cuza” University of Iasi, Carol I Avenue, 20A, 700505 Iași, Romania; andracurpan@yahoo.com (A.-S.C.); lungufabian123@gmail.com (P.F.L.); 3Internal Medicine Clinic, University of Life Sciences “Ion Ionescu de la Brad”, Mihail Sadoveanu Street, No. 3, 700490 Iasi, Romania; lumidih@yahoo.com; 4Faculty of Veterinary Medicine, University of Life Sciences “Ion Ionescu de la Brad”, Mihail Sadoveanu Street, No. 3, 700490 Iasi, Romania; buzenchi.teodora@yahoo.com (T.M.B.P.); gsolcan@uaiasi.ro (G.S.); csolcan@uaiasi.ro (C.S.); 5Department of Exact Sciences and Natural Sciences, Institute of Interdisciplinary Research, “Alexandru Ioan Cuza” University of Iasi, Carol I Avenue, 20A, 700505 Iași, Romania; balmus.ioanamiruna@yahoo.com; 6Research Center on Advanced Materials and Technologies, Department of Exact and Natural Sciences, Institute of Interdisciplinary Research, “Alexandru Ioan Cuza” University of Iasi, Carol I Avenue, 20A, 700505 Iași, Romania; roxanajijie@yahoo.com; 7Department of Biology, Faculty of Biology, “Alexandru Ioan Cuza” University of Iasi, Carol I Avenue, 20A, 700505 Iași, Romania; alin.ciobica@uaic.ro; 8Academy of Romanian Scientists, 3 Ilfov, 050044 Bucharest, Romania; 9Center of Biomedical Research, Romanian Academy, Iasi Branch, Teodor Codrescu 2, 700481 Iasi, Romania; 10Preclinical Department, Apollonia University, 700511 Iasi, Romania

**Keywords:** microplastics, zebrafish, oxidative stress, immunohistochemistry, schizophrenia

## Abstract

The effects of exposure to environmental pollutants on neurological processes are of increasing concern due to their potential to induce oxidative stress and neurotoxicity. Considering that many industries are currently using different types of plastics as raw materials, packaging, or distribution pipes, microplastics (MPs) have become one of the biggest threats to the environment and human health. These consequences have led to the need to raise the awareness regarding MPs negative neurological effects and implication in neuropsychiatric pathologies, such as schizophrenia. The study aims to use three zebrafish models of schizophrenia obtained by exposure to ketamine (Ket), methionine (Met), and their combination to investigate the effects of MP exposure on various nervous system structures and the possible interactions with oxidative stress. The results showed that MPs can interact with ketamine and methionine, increasing the severity and frequency of optic tectum lesions, while co-exposure (MP+Met+Ket) resulted in attenuated effects. Regarding oxidative status, we found that all exposure formulations led to oxidative stress, changes in antioxidant defense mechanisms, or compensatory responses to oxidative damage. Met exposure induced structural changes such as necrosis and edema, while paradoxically activating periventricular cell proliferation. Taken together, these findings highlight the complex interplay between environmental pollutants and neurotoxicants in modulating neurotoxicity.

## 1. Introduction

Schizophrenia (SZ) is a chronic psychiatric disorder with a global prevalence of approximately 0.4% [1]. This disorder is generally characterized by the presence of positive symptoms (delusions, hallucinations), cognitive symptoms (disorganized speech and thought processes) [2], negative symptoms (reduced emotional expression, inability to start or continue tasks, anhedonia, alogia) [2,3], or a combination of the above. The presence of negative symptoms is correlated with a significant segment of the comorbidities associated with SZ [2]. The onset of SZ usually occurs early during neurogenesis, well before cognitive development, meaning that the development of the social skills needed in adulthood could be impaired [4,5]. The poor prognosis of SZ is associated with early onset, the male sex, occurrence of negative symptoms, a family history of SZ, social isolation, adverse childhood experiences, and a lack of affection [6].

Zebrafish has become a reliable species for modeling schizophrenia-like symptoms, anxiety, depression, and neurodegenerative disorders [7], especially through pharmaceutical manipulation [8]. In the context of the present study, one of the most popular approaches for modeling schizophrenia is the administration of ketamine [9] for mimicking the positive symptoms [10], methionine for the negative symptoms [11], and their combined effect for a more robust model that includes the cognitive dysfunction as well [12].

Diet plays a crucial role in psychiatric disorders due to its impact on brain function and overall mental health; it is worth mentioning that this is an often-missed component in SZ management. The proper function of the brain requires a variety of nutrients, including vitamins, minerals, omega-3 fatty acids, and amino acids [13,14]. Thus, it was shown that deficiencies of certain nutrients could contribute to the development or exacerbation of psychiatric symptoms. Also, while chronic inflammation was associated with several psychiatric disorders, it was recently reported that diets rich in processed food, refined sugars, and unhealthy fats could promote inflammation [15]. On the other hand, diets mainly based on fruits, vegetables, whole grains, and omega-3 fatty acid intake could exhibit anti-inflammatory effects [15].

Emerging research has suggested a strong connection between the brain and the gut, known as the brain–gut axis. The gut microbiota, consisting of trillions of microorganisms living in the gastrointestinal tract, was demonstrated to modulate several brain functions and behavior via the production of neurotransmitters and immune regulators [16,17].

A growing concern of modern society is related to the widespread presence of PM in various environments, even reaching food sources [18]. MPs can contaminate food sources via air, water, and soil contamination, as well as via packaging and processing [19,20]. While the effects of MP-contaminated food are not currently fully described, there are some concerns about the potential impact of MPs on human health [21]. Additionally, MPs were recently detected in human’s samples such as thrombi [22] or, even worse, in semen samples [23]. This is an even clearer indication of the fact that people are extraordinarily exposed to this type of pollutant. MPs can be mixed with additives and other pollutants that were previously linked to inflammatory response when entering a living body [24], or cyanobacteria, if we consider seafood, where recent studies indicate that co-exposure of Microcystin-LR, an natural toxin produced by cyanobacteria, lead to an increased MDA level along with hepatic histopathological injuries [25] Thus, the intake of MP-contaminated food could indirectly affect psychiatric health by influencing gut microbiota composition and function [20,26]. MPs have been shown to induce oxidative stress, leading to cellular damage and inflammation [27,28,29]. Oxidative stress is an important pathophysiological component of psychiatric disorders, having the potential to exacerbate or aggravate some of the symptoms [30,31,32,33], as well as to actively contribute to the development of several neuropsychiatric diseases (i.e., SZ, Alzheimer’s disease, Parkinson’s disease, and major depressive disorder) [34,35,36,37,38,39].

According to several studies conducted in zebrafish models, MPs can pass through the blood–brain barrier [38], accumulating at high levels in nervous system structures and leading to a considerable increase in oxidative imbalance and apoptosis [39,40]. In addition, SZ patients have been shown to exhibit increased levels of lipid peroxidation and protein oxidation in the blood, cerebrospinal fluid, and tissues, along with altered antioxidants and reduced glutathione [40,41,42].

In the present study, we experimentally induced schizophrenia-like symptoms in zebrafish using methionine, ketamine, and a combination of methionine and ketamine, along with exposure to microplastics. Thus, we aimed to study the contribution of contaminated food in the development of oxidative imbalance, with special attention to the systemic and tissue status in the context of well-established models of SZ in zebrafish based on methionine (Met) and ketamine administration (Ket), as previously described by our group [12].

## 2. Results

### 2.1. Oxidative Stress Levels

The analysis of biochemical data revealed several effects of MPs and pharmacological agents. We observed that the administration of Ket, Met, and their combination did not lead to significantly increased SOD specific activity compared to the control group. On the other hand, only MP administration in MP+Ket led to a decrease in SOD specific activity compared to the Ket group (*p* = 0.018) (Figure 1A, Table 1). Other significant variations in the specific enzymatic activity of SOD were recorded for Met vs. MP+Ket (*p* = 0.046), Met+Ket vs. MP+Ket (*p* = 0.049), and MP+Ket vs. MP+Met (*p* = 0.005).

Regarding GPx specific activity (Figure 1B, Table 1), we observed significant differences between the following groups: CTR vs. MP+Met+Ket (*p* < 0.001), MP vs. MP+Met+Ket (*p =* 0.005), Ket vs. MP+Met+Ket (*p =* 0.041), MP+Ket vs. MP+Met+Ket (*p* < 0.01), and MP+Met vs. MP+Met+Ket (*p =* 0.01), MP+Ket vs. MP+ Met+ KET (*p =* 0.003). When strictly compared with the control group, all groups had higher levels of GPx except MP and MP+Ket, indicating different degrees of antioxidant defense mechanisms and a compensatory mechanism.

We found significant variations in the relative content of MDA (Figure 1C) when compared with the control group in two treatments, namely MP (*p* = 0.006) and MP+Ket (*p* = 0.0211), indicating increased oxidative stress. Moreover, significant differences occurred for MP vs. MP+Met, with a significant decrease in the MDA content (*p* = 0.002), as well as for MP vs. MP+Met+Ket (*p* = 0.022). Other significant differences were found also in the following combinations: Ket vs. MP+Met (*p* = 0.028) and MP+Ket vs. MP+Met (*p* = 0.012) (Table 1).

### 2.2. Immunohistology Results

Several histopathological changes were observed in the different layers of the optic tectum in zebrafish following exposure to microplastics, ketamine, methionine, and their combination (Figure 2, Table 2). The severity and frequency of lesions in these layers were more pronounced when combining MP+Ket and MP+Met and more attenuated when combining MP+Met+Ket and Met+Ket.

The optical tectum showed marked structural differences in fish from all experimental groups. Areas of granular cell necrosis and detachment in the SPV area, granular cell spongiosis in SFGC, small areas of necrosis in the outer layer, connective tissue degeneration, edema, and elevated and degenerated external squamous cells were observed. These changes were more pronounced in individuals exposed to the combination of microplastics and ketamine and included detachment and necrosis in the granular cells in the SPV area and necrosis in mononuclear cells, as well as the high detachment and degeneration of neuronal cells in the SO and MS areas.

The pathology was also pronounced in fish exposed to ketamine and methionine, and observations included the spongiosis of granular cells in the SPV area and vacuolation in granular cells in the SAC layer due to degeneration. A large number of blood capillaries in the SFGC area and a reduction in the SO and MS layers due to neuronal cell degeneration were also observed. In the periventricular area, ectasia of the vessels, a variable proliferation of periventricular stem cells, and edema were observed.

PCNA expression varies by group. The number of positively labeled cells was recorded, in descending order, in the following groups: MP, MP+Met, Met, Ket, MP+Ket, MP+Met+Ket, and Met+Ket. In the control group, positive PCNA cells are present, in moderate amounts, at the level of the regeneration areas.

MPs and Met have an important role in stimulating the multiplication of progenitor stem cells or radial glial cells (RGCs) in the periventricular zone, optic tectum, and valvula cerebeli. In all experimental groups, both MPs, Met, and Ket alone and combined produce many changes that are preceded by stimulation of neurogenesis. Thus, the presence of positive PCNA cells can be explained, both at the level of the regeneration niches and at those in migration, in groups with MP, MP+Ket, and MP+Met (Figure 2).

In the groups receiving ketamine, PCNA positive cells were observed in the two areas of the periventricular and cerebellar valve, but a smaller number of migratory cells was also noted. It is possible that ketamine treatment not only alters the expression pattern of *PCNA* but also the time of migration/differentiation of cerebellar cell types in adults, contributing to a state of general alteration of CNS cell proliferation.

Regarding methionine, it caused structural changes in all groups that were exposed to the agents, either alone or in combination, and generated areas of necrosis in the ZPV, edema, and blood vessels and also an activation of periventricular neurogenesis with *PCNA* positive cells. *PCNA* expression varied by group. In the SPV area, the number of positively labeled cells was recorded, in descending order, as follows: MP+Met, Met, Ket, MP, MP+Met+Ket, control, Met+Ket, and MP+Ket (Table 2). In the TO, the number of positively labeled cells was recorded, in descending order, as follows: MP, MP+Met, Met+Ket, and MP+Met+Ket and there was a more reduced positivity in Ket, Met, and Met+Ket (Table 2). In the control group, 1–30 positive *PCNA* cells/field were present at the level of the regeneration areas.

*TNFαI8* registers the most intense expression in the periventricular area and in the optical tectum and, subsequently, varies by group. In the SPV area, the number of positively labeled cells was recorded, in descending order, as follows: MP+Met, MP+Met+Ket, MP, Met+Ket, Met, Ket, and control. In TO, the number of positively labeled cells for TNFAI8 expression was recorded in descending order, as follows: Met, MP, MP+Met+Ket, Ket, Met+Ket, MP+Met, MP+Ket, and control (Table 2). The presence of *TNFαI8* expression in the nervous system of fish from experimental groups denotes the presence of an inflammatory process generated by MP, Ket, Met, and their combination (Figure 2).

The expression of *Cox41-i* was also recorded. In the SPV area, the number of positively labeled cells for *Cox41-I* was recorded, in descending order, as follows: MP+Met+Ket, KET, MP, MP+Ket, MP+Met, Met, Met+Ket, and control (Table 2). In the TO, the number of positively labeled cells for *Cox41-i* was recorded, in descending order, as follows: MP+Ket, Ket, MP+Met, MP, MP+Met+Ket, Ket, Met+Ket, and control (Table 2).

The presence of a different *Cox41-i* expression in cells in the SPV zone and the TO denotes different degrees of ATP generation, dependent on the expression of hypoxia induction factor (Figure 2). The *Cox41-i* overexpression in some groups indicates increased energy requirements to restore the cellular metabolism of nerve cells as a result of the oxidative stress and neurotoxicity produced by types of various exposure.

*BDNF 2* expression in the SPV area was recorded, in descending order, as follows: control, MP+Met +Ket, MP+Met, MP, Ket, Met, MP+Ket, and Met+Ket. In the TO, the number of positively labeled cells was recorded, in descending order, as follows: MP+Met, Met, MP+Met+Ket, control, MP, MP+Ket, Ket, and Met+Ket (Figure 3, Table 3).

*MAP2* expression in the SPV area was recorded, in descending order, in the following groups: Met, MP+Met+Ket, Ket, control, MP, Met +Ket, MP+Met, and MP+Ket. In the TO, *MAP2* expression was recorded, in descending order, as follows: Met, MP+Met+Ket, MP+Ket, Ket, MP+Met, control, MP, and Met+Ket (Figure 3, Table 3).

*H2A* expression in the SPV area was recorded, in descending order, as follows: MP+Met+Ket, Met+Ket, MP+Ket, Met, Ket, MP, MP+Met, and control. In the TO, *H2A* expression was recorded, in descending order, as follows: MP+Ket, Met, Met+Ket, MP+Met+Ket, MP, Ket, MP+Met, and control (Figure 3, Table 3).

The expression of *S100* is intense in all groups (Figure 4, Table 4). Because the *S100* protein continues to be expressed in these regions of adult zebrafish and by subventricular zone glial cells, the *S100* protein plays an important role in the neurogenesis of adult zebrafish. In the SPV area, *S100* expression was recorded, in descending order, as follows: MP, MP+Ket, MP+Met+Ket, Ket, MP+Ket, Met, Met+Ket, and control. In the TO, *S100* expression was recorded, in descending order, as follows: MP+Met, Ket, MP, Met+Ket, MP+Ket, Met, MP+Met+Ket, and control (Figure 4, Table 4).

The expression of *GFAP* varies, including having the lowest intensity of expression in the control group and the most intense expression in the MP group. In the SPV area, *GFAP* expression was recorded, in descending order, as follows: MP, MP+Ket, MP+Met, Ket, MP+Met+Ket, Met+Ket, Met, and control. In the TO, the expression of GFAP descended in the following order: MP+Ket, MP, Ket, Met, Met+Ket, MP+Met, MP+Met+Ket, and control (Figure 4, Table 4). The changes produced by MP, Met, and Ket alone and combined caused a reaction in astrocytes but without producing scar areas as in mammals.

*Tub2* expression was observed in periventricular cells in all experimental groups but was lower in the control. In the SPV area, the expression of *Tub2* is more intense in the MP+Ket group, descending in the following order: MP+Ket, MP+Met, Met+Ket, MP, MP+Met+Ket, Ket, control, and Met. In the TO area, *Tub2* expression was recorded, in descending order, as follows: MP+Met, MP+Met+Ket, MP, MP+Ket, Met+Ket, Ket, control, and Met (Figure 4, Table 4).

## 3. Discussion

This study demonstrated effects on oxidative stress and immunohistochemistry (IHC) for microplastics alone and in three Schizophrenia models, focusing on an already vulnerable population category; this approach represents an important step forward to fill the gaps in this field of molecular toxicology.

In terms of oxidative stress, SOD is an antioxidant enzyme that plays a crucial role in neutralizing superoxide radicals. High levels of SOD activity suggest an adaptive mechanism to counteract increased oxidative stress and protect cells from potential damage [43]. On the other hand, decreased SOD activity could suggest a drop in this enzyme’s ability to neutralize superoxide radicals, which could lead to the increased vulnerability of cells to oxidative stress. These effects could be seen in locations when the balance between the production of reactive oxygen species and the antioxidant defense performance is disrupted [43]. Previous studies have shown that 28-day chronic exposure to polypropylene, both alone and co-exposed with triclosan, could lead to antioxidant defense weakening via decreasing the neutralizing activity of SOD [44,45]. In these situations, increased MDA production was often observed [45]. High MDA levels could be associated with increased oxidative stress and could at least explain the occurrence of certain pathological processes, including inflammation, cardiovascular disease, and neurodegenerative disorders [46].

In a study designed to assess the transgenerational effects of methionine exposure, it was found that MDA levels show decreases over the control group, whereas SOD activity increased in the f1 generation [47]. Notably, a significant decrease in SOD activity was observed after cessation of Met exposure [48]. With regards to the effects of Ket administration, the previous studies indicated that the levels of SOD and MDA were increased in a dose-dependent manner even after Ket administration discontinuation [49]. In these contexts, it was suggested that both Met and Ket could alter the oxidative balance, influence the antioxidant enzymatic defense performance, and lead to the occurrence of cellular damage, in a similar manner to the results of our current study.

GPx is another member of the antioxidant enzymatic defense system that is implicated in cellular protection against the harmful effects of ROS overproduction. Its main action is to catalyze the neutralization of hydrogen peroxide. However, in some situations, GPx can also use other peroxides as enzymatic substrates, leading to their neutralization (including lipid peroxides) [50]. Our results showed that the exposure to MPs alone did not lead to altered GPx antioxidant activity; this was in contrast to [29], which showed that 21-day exposure to a cocktail of different microplastics (1 mg L^−1^) led to significantly lower GPx activity as compared to the control group. On the other hand, we observed decreased GPx activity when MP and Ket were co-administered, meaning a reduced ability to neutralize ROS and increased cellular susceptibility to oxidative damage.

In contrast to other studies that reported no changes in GPx activity when Met and Ket were independently administered [51,52], we found that the co-administration of MP, Met, and Ket led to significantly increased GPx activity. As the effects of Met and Ket on the dynamics of GPx activity were similar to those in the already mentioned studies [51,52], the different patterns of variation we observed for the MP+Met+Ket co-administration could suggest that the fish metabolism deployed counteractive measures against oxidative stress by modulating defense mechanisms. This response is often observed in various conditions, including inflammation, the immune response to pathogens, or upon exposure to environmental stressors [53].

Plastic pollution is a critical worldwide environmental concern and comprises different sizes of plastic particles, categorized by the scientific community as macroplastics, microplastics, and nanoplastics [54]. MPs are currently defined as small plastic fragments of various shapes and sizes (ranging between 100 nm and 5 mm) and are considered to be an ecological danger that requires the utmost attention due to their presence in aquatic ecosystems and their toxicity for all living organisms [55]. Moreover, the association of MPs with other substances, such as pharmaceutical products and byproducts, has become of great interest to researchers [56,57,58,59].

Multiple studies have shown that MPs, alone or in combination with heavy metals, could exhibit detrimental effects on the neurological functions of fish, such as acetylcholinesterase activity inhibition [60,61] or neurotransmitter activity alterations [28,62,63,64]. Moreover, studies have reported on the negative effects of MPs in zebrafish during different developmental stages, especially early neurogenesis impairment, which could lead to persistent and severe alterations of morphology and locomotion, as well as feeding and predator avoidance behavior [61,65,66,67]. Furthermore, the multi-generational maternal and paternal transfer of nanoplastics was previously documented in the *Danio rerio* species [68].

Regarding the potential of MPs to modulate ROS production within the brains of fish, several studies have reported that this effect could be observed regardless of the size of the MPs (nano, micro, and macro) [69,70,71]. In this context, the over-production of ROS within the tissues of the central nervous system was previously associated with decreased cognitive performances and weakened antioxidant defense [72].

Previous studies on the effects of MPs revealed that a concentration of 100 mg L^−1^ of MP in tilapia and carp is sufficient to produce several histopathological changes, including necrosis, spongiosis, degeneration, and oedema [73,74]. Similar effects were reported in goldfish (*Carassius auratus*) exposed to pure MP [75] and zebrafish larvae (*Danio rerio*) exposed to pure low-density polyethylene fragments [76]. Exposure to polypropylene MP in concentrations of 300 mg L^−1^ and 600 mg L^−1^ lead to downregulated pathways and genes associated with cell proliferation regulation and DNA damage repair mechanisms in the liver, highlighting the need for future studies regarding PP toxicity [77]. Savari et al. in 2020 analyzed the brain tissues of *Epinephelus coioides* and reported hyperemia, hemorrhage, karyosis, tissue necrosis, hyper chromatin, vacuolation, endothelial hypertrophy, dropsy degeneration, and granular ectopic accumulation following the exposure to methylmercury [78]. These findings suggest that MP exposure could lead to similar effects to those observed in the case of heavy metal intoxication.

The optic tectum is a bilobate structure located in the dorsal part of the midbrain. This region of the central nervous system (CNS) mainly acts as a visual center and consists of six different layers with neurons of different shapes and sizes. In contrast to mammals, adult zebrafish have a high number of neurogenic niches distributed throughout the brain with a high capacity for regeneration without the formation of scars or injuries that contribute to visual information processing [79,80]. Also in zebrafish, these highly proliferative areas are widespread and can be detected in all subdivisions of the brain, including the telencephalon, diencephalon, midbrain, and metencephalon [81]. Significant levels of proliferation could be found, especially in the thalamus and specifically in the regions surrounding the habenula, near the optic tectum, along with its subdivisions [80,81]. Adolf et al. (2006) have shown via IHC staining with proliferating cell nuclear antigen (*PCNA*) that the telencephalon contains two different types of neural precursor cells: (1) slow-cycling ones, distributed along the ventricular surface, and (2) fast-cycling ones, mainly organized in a sub palatial group [82]. Slow-cycling progenitors have been identified as radial glial cells (RGCs), while rapidly multiplying progenitors have been described as neuroblasts [81].

Cell proliferation and neurogenesis are two processes susceptible to environmental pollution that have negative consequences for the body. Some studies mention that PCNA-positive cells and cell proliferation themselves decrease in zebrafish brains after exposure at very young ages to Cu and Cu+MP, consistent with reduced *PCNA* gene expression [61]. This inhibition of proliferation and the reduction in *PCNA* expression was recorded in the Ket+Met group, suggesting that this combination could be responsible for oxidative stress, changes in protein phosphorylation, inhibition of G1 to S transition during mitosis, and dysregulation of intracellular calcium ion (Ca^2+^) homeostasis, as previously documented [82].

It has been shown that between 6 months and 2 years of age, the pool of neural stem cells and progenitor cells gradually depletes, which impacts neuronal regeneration, especially in adults [83]. Thus, despite the observation of cyclic cells in the brains of 7-month-old zebrafish via IHC, the lack of detection of *PCNA* via WB agrees with the findings in the literature, indicating a decrease in the population of NSC cells and cyclic cells in the CNS of zebrafish that were older than 6 months, which makes quantification more difficult.

Our findings showed that Ket administration altered autophagy, as well as proliferation. Researchers have suggested that Ket-induced neurotoxicity could be correlated with increased levels of ROS [84]. In addition, Ket increased ROS production alongside the cell death and differential expression of oxidative-stress-related genes in human neurons in a time- and dose-dependent manner [85]. It is recognized that frequent Ket administration reduces the expression of autophagy proteins, such as LC3, in rat models of traumatic brain injury, probably by activating the mTOR signaling pathway [86], and indeed, Ket has been described as activating mTOR signaling in mammals [84], which is a key regulator of cell growth and proliferation. These results support the disruption of neuronal proliferative and autophagic processes by Ket, as reported in other species and described in humans.

*BDNF* could act as an antioxidant factor, as it is known to increase the activity levels of some antioxidant enzymes (i.e., SOD and GPx) [87]. The presence of intense *BDNF* expression in the experimental groups demonstrated the ability to activate the antioxidant enzyme system to restore damage caused by MP, Met, Ket, and their combination. The presence of *MAP2* in dendritic extensions emphasizes their role in stabilizing microtubules and is characteristic of differentiated neurons. In addition, immunostaining revealed that *H2A.z* was ubiquitously expressed in the developing cortex and was predominantly localized to the nucleus in the VZ/SVZ (indicating *H2A.z* expression in neuronal progenitor cells), suggesting that *H2A.z* is required for cortical neurogenesis. Thus, our study provides further potential evidence to identify *H2A.z* as a risk gene for neurodevelopmental disorders.

Some significant observations of the current study are the fact that the PCNA and BDNF expression in the MP groups increased, which suggests the presence of DNA lesions and a possible cognitive decline, whereas in terms of inflammation, it was present in all groups besides the control. Moreover, *MAP2* expression has been associated with schizophrenia, which in the context of our study supports the animal models used. In terms of the connection between the immunohistology findings and oxidative stress, *Cox41-i* expression is responsible for the production of superoxide species, whereas tumor necrosis factor interleukin-8 (*TNFIL-8*) is responsible for inducing oxidative stress by producing reactive oxygen species (ROS) and pro-inflammatory cytokines.

Broadly speaking, the results of our current study contribute to filling the knowledge gap within the topic of the neurotoxicity risks of MPs to aquatic organisms. The biochemical and immunohistological responses are consistent with the neurotoxic effects observed in fish treated with MP, Ket, Met, and their combinations. Moreover, alterations in fish exposed to microplastics are consistent with those observed in fish treated with other environmental toxicants, such as heavy metals and metal nanoparticles. It was also observed that the alteration of the evaluated biomarkers was dependent on the exposure design (with special regards to the co-exposure groups), which could lead to further criteria for biological responses to MP exposure, especially in the context of Schizophrenia disorder models. The effect of MP size may be related to the ability to penetrate cells and tissues and their potential to induce molecular damage due to their physical and chemical properties.

## 4. Materials and Methods

### 4.1. Ethical Statement

All experimental procedures were approved by the Ethics Committee of the Faculty of Biology, “Alexandru Ioan Cuza” University of Iasi (no. 2533/08.09.2022) and the Ethics Committee of the Faculty of Veterinary Medicine, University of Agricultural Sciences and Veterinary Medicine “Ion Ionescu de la Brad”, Iasi (no. 165/26.01.2022) and were conducted in agreement with the European Directive and Romanian Legislation (2010/63/EU and 43/2014) regarding the protection of animals used for scientific or experimental purposes.

### 4.2. Animals and Housing

A total of 80 wild-type adult zebrafish obtained from an authorized local source were housed according to the literature guidelines (28 °C, 10 L tanks with oxygen pumps, light/dark cycle of 12:12 h, and fed once a day with commercial food for fish). To eliminate any possibility of interference with our experiments and animal welfare, we tested the water parameters daily, measuring nitrites, nitrates, water hardness, alkalinity, pH, and chloride.

### 4.3. Experimental Models of Schizophrenia and MPs Pollution

After the acclimatization period (14 days), the batch of zebrafish was randomly assigned to eight experimental groups (n = 10/group), the control group (CTR), ketamine group (Ket), methionine group (Met), ketamine+methionine group (Met+Ket), microplastics control (MP), ketamine + MP group (MP+Ket), methionine + MP group (MP+Met), and the co-exposure group (MP+Met+Ket).

Since the current report is part of a more complex study, the experimental design was performed according to the previously published protocol [12]. Briefly, Ket and Met were chronically administrated (5 min exposure to 0.1% Ket/day in separate individual tanks for 5 consecutive days; the tank water containing extemporaneously prepared 6.0 mM Met, replaced every 24 h for 7 consecutive days), while MPs (polypropylene microfibers, <2 mm size) were administered at a concentration of 2 mg L^−1^ at the same time as the food, in an amount required based on the weight of the fish, for 7 consecutive days [88]) for a period of 7 days. The co-exposure protocol consisted of delayed Ket administration (48 h after MP and Met administration).

### 4.4. Oxidative Stress Analysis

After the exposure protocol was performed, all the animals were euthanized according to European guidelines for standard euthanasia procedures by immersion in cold water for at least 10 min or until opercular movements ceased. The whole bodies were rinsed with distilled water and underwent mechanical processing. The homogenate was mixed with phosphate-buffered saline (PBS) solution and centrifuged at 3500 rpm for 15 min. The fresh supernatant was collected, aliquoted, and used to determine the enzymatic activities of superoxide dismutase (SOD) and glutathione peroxidase (GPx) and the malondialdehyde (MDA) content.

Superoxide dismutase (SOD) activity was determined via an indirect method based on Dojindo’s tetrazolium salt (WST-1) reaction with a superoxide anion according to the manufacturer’s analytical kit protocols (SOD Assay Kit, Sigma, Roedermark, Germany).

Likewise, GPx activity was determined according to the manufacturer’s protocols (GPx Cellular Activity Assay Kit CGP-1, Sigma, Germany) via an indirect method of dynamic observation of substrate consumption, where the rate of NADPH consumption during the considered time unit was used as an indicator of GPx activity.

Malondialdehyde (MDA), which is a measure of lipid peroxidation, was spectrophotometrically measured at 532 nm in a UV–VIS spectrum (Analytik Jena Specord 200, Germany). The MDA content was determined using the thiobarbituric acid reactive substances (TBARS) method, according to the previously described protocols [89,90]. The concentrations of MDA were expressed as relative content per mg of tissue (μmol MDA/mg tissue).

The total soluble protein (TSP) content was assessed based on the Bradford method using a commercial kit according to the manufacturer protocols (Rapid Protein Quantification Kit, Sigma, Germany). The results obtained were calculated against a bovine serum albumin etalon curve and were used to express the antioxidant enzyme activity as specific activity (EU/mg TSP) [89,90]

### 4.5. Statistical Analysis

All the numerical analyses were performed using Graph Pad Prism software version 9 (San Diego, CA, USA). Firstly, the normality and distribution of the data were determined using the Shapiro–Wilk test. Since the normality test was passed, a two-tailed Student’s *t* test was applied for comparisons between the groups. The data are expressed as the average ± standard error of the mean (SEM), and a *p* < 0.05 was considered to be statistically significant.

### 4.6. Immunohistology Assays

The fish heads were collected shortly after the fish were euthanized, fixed with Bouin for 48 h and decalcified with EDTA 15% for 7 days, followed by dehydration with ethylic alcohol 70–99.8% and clarification with xylene and fixing in paraffin. The samples were then sectioned at 4 µm, stained with hematoxylin plus eosin (H&E), and immunohistochemically (IHC) colored with the following antibodies: *PCNA*, *TNFAI8*, *Cox41*, *BDNF*, *Map2*, *H2A*, *S100*, *GFAP*, and *Tub2* (Table 5). The sectioning was performed longitudinally.

For each antibody, the number of immunolabeled cells was counted on five fields at magnitude of ×400. The score for immunopositivity was established as follows: + = 1–15 positive cells; ++ = 15–30 positive cells; +++ = 30–60 positive cells; ++++ = 60–100 positive cells.

## 5. Conclusions

Considering the impact that microplastics pollution may have on health and overall wellbeing, the present study proposed an experimental approach measuring the effect of MPs on certain a psychiatric condition, namely schizophrenia. This approach represents an important step forward as it focuses on an already vulnerable category of people and the importance of a clean diet. The presence of MPs alone can alter the measured biomarkers, indicating the potential toxicity of microplastics even without additional contaminants. The severity and frequency of lesions in the optic tectum are more pronounced when MPs are combined with ketamine, as well as with methionine. MPs and methionine seem to play a significant role in stimulating the multiplication of progenitor stem cells or radial glial cells in areas such as the periventricular region, optic tectum, and cerebellar valve. Moreover, methionine exposure leads to structural changes, including creating areas of necrosis in the periventricular zone, across all exposed groups, whether alone or in combination. Despite these detrimental effects, methionine also activated periventricular neurogenesis, as evidenced by the presence of PCNA-positive cells. Moreover, we observed that MPs could have protective effects against oxidative stress, as their combined exposure with Met and Met+Ket led to a successful second antioxidant line of defense alongside reduced lipid peroxidation. Overall, the findings suggest complex interactions between microplastics, ketamine, and methionine in the context of neurogenesis and optic tectum lesions. Future studies will be necessary for a more complete understanding of the mechanisms of action.

## Figures and Tables

**Figure 1 ijms-25-08331-f001:**
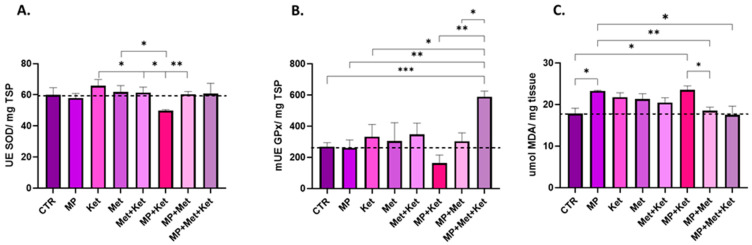
The graphic representation of the oxidative stress markers levels. (**A**)—SOD specific activity (UE SOD/mg TSP), (**B**)—GPx specific activity (mUE GPx/mg TSP), (**C**)—relative MDA content (μmol MDA/mg tissue). All data are expressed as means ± SEM (n = 5/per group, * *p* < 0.05; ** *p* < 0.01; *** *p* < 0.001, two-tailed student *t* test). TSP = total soluble protein content.

**Figure 2 ijms-25-08331-f002:**
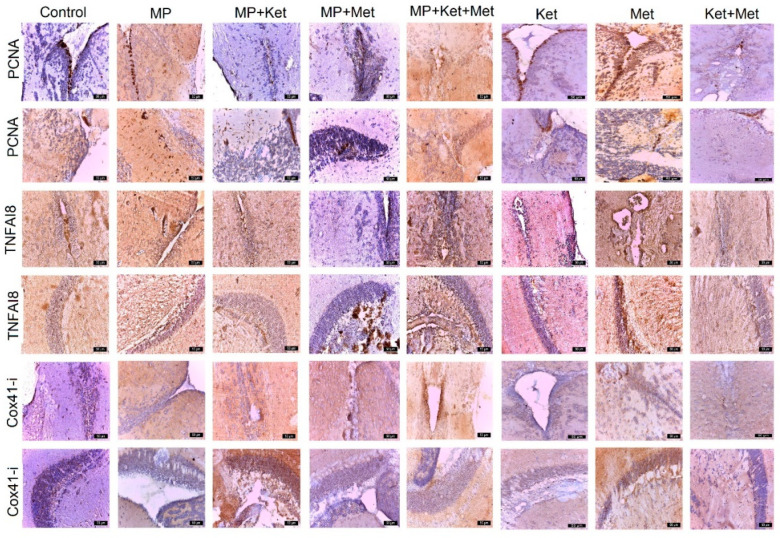
Different expression of *PCNA*, *TNFAI8*, and *Cox41-i* in different experimental and control groups. In rows 1, 3, and 5 the periventricular expression is observed, and in rows 2, 4, and 6, the expression is observed at the level of the optic tectum (bar = 50 µm).

**Figure 3 ijms-25-08331-f003:**
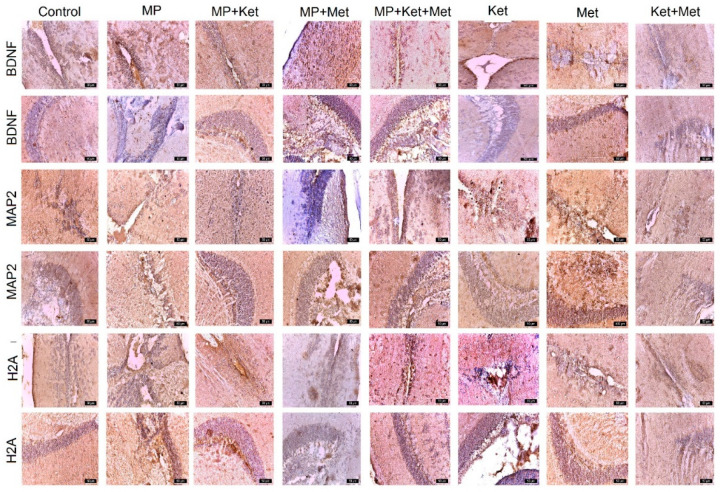
Different expression of *BDNF*, *MAP2*, and *H2A* in different experimental groups and controls. In rows 1, 3, and 5, the periventricular expression is observed, and in rows 2, 4, and 6, the expression is observed at the level of the optic tectum (bar = 50 µm).

**Figure 4 ijms-25-08331-f004:**
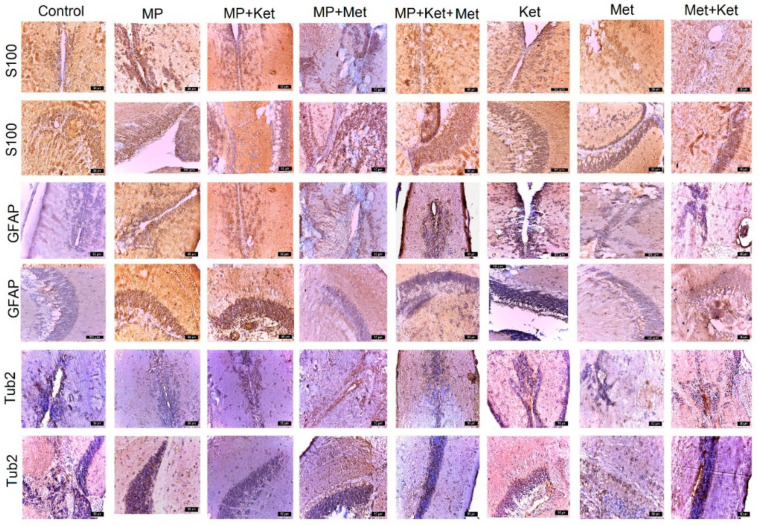
Different expression of *S100*, *GFAP*, and *Tub2* in different experimental and control groups. In rows 1, 3, and 5, the periventricular expression is observed, and in rows 2, 4, and 6, the expression is observed at the level of the optic tectum (bar = 50 µm).

**Table 1 ijms-25-08331-t001:** Oxidative stress marker variation, depending on the administered treatment.

	Experimental Group			Oxidative Stress Parameters ^‡^		
	MP	Ket	Met	SOD	MDA	GPx
1	+	−	−	↓	↑ **	−
2	−	+	−	↑	↑	↑
3	−	−	+	↑	↑	↑
4	−	+	+	↑	↑	↑
5	+	+	−	↓	↑ ^*^	↓
6	+	−	+	−	↓	↑
7	+	+	+	−	−	↑ ***

+ substance administered; − substance not administered; ‡ the control group was used as reference; */**/*** the power of significance between the treatment and the control groups (* *p* < 0.05; ** *p* < 0.01; *** *p* < 0.001).

**Table 2 ijms-25-08331-t002:** Score of expression of PCNA, TNFAI8, and *Cox41-i* in different experimental and control groups (number of positive immunolabeled cells/field at a magnitude of ×400).

Antibody	Control	MP	MP+Ket	MP+Met	MP+Met+Ket	Ket	Met	Met+Ket
*PCNA* SPV	++	++	+	+++	++	+++	+++	+
*PCNA* TO	++	+++	++	++	++	+	+	+
*TNFAI8* SPV	+	+++	++++	++	++++	++	++	++
*TNFAI8* TO	+	+++	++	++	+++	++	++++	++
*Cox41-i* SPV	++	+++	+++	+++	++++	++++	++	++
*Cox 41-i* TO	+	+++	++++	+++	++	++++	++	++

SPV = periventricular space; TO = optic tectum; + = 1–15 positive cells; ++ = 15–30 positive cells; +++ = 30–60 positive cells; ++++ = 60–100 positive cells.

**Table 3 ijms-25-08331-t003:** Score for the expression of BDNF, MAP2, and H2A in different experimental groups and the control (number of positive immunolabeled cells/field at a magnitude of ×400).

Antibody	Control	MP	MP+Ket	MP+Met	MP+Met+Ket	Ket	Met	Met+Ket
*BDNF* SPV	++++	+++	++	+++	++++	+++	+++	++
*BDNF* TO	++	++	++	++	++	+	+++	+
*MAP2* SPV	+++	+++	++	+++	++++	+++	++++	+++
*MAP2* TO	++	++	+++	++	+++	+++	++++	+
*H2A* SPV	++	+++	+++	++	++++	+++	+++	++++
*H2A* TO	++	+++	+++	++	+++	++	+++	+++

SPV = periventricular space; TO = optic tectum; + =1–15 positive cells; ++ = 15–30 positive cells; +++ = 30–60 positive cells; ++++ = 60–100 positive cells.

**Table 4 ijms-25-08331-t004:** Score of expression of expression of S100, GFAP, and Tub2 in different experimental and control groups (number of positive immunolabeled cells/field at a magnitude of ×400).

Antibody	Control	MP	MP+Ket	MP+Met	MP+Met+Ket	Ket	Met	Met+Ket
*S100* SPV	+++	++++	++++	+++	++++	+++	+++	+++
*S100* TO	+++	++++	+++	++++	+++	++++	+++	+++
*GFAP* SPV	+	++++	+++	++++	+++	++++	++	+++
*GFAP* TO	+	++++	++++	++	+	++	++	++
*Tub2* SPV	+++	++	+++	+++	+++	++	+	+++
*Tub2* TO	++	++	++	+++	++	++	+	++

SPV = periventricular space; TO = optic tectum; + = 1–15 positive cells; ++ = 15–30 positive cells; +++ = 30–60 positive cells; ++++ = 60–100 positive cells.

**Table 5 ijms-25-08331-t005:** Primary and secondary antibodies alongside the originating species and the applied dilution.

**Primary Antibody**	**Species for Primary Antibody**	**Primary Antibody Dilution**	**Species for Secondary Antibody**	**Secondary Antibody Dilution**
** *PCNA* **	Rabbit	1:250	Goat anti rabbit	1:250
** *TNFAI8* **	Rabbit	1:150	Goat anti rabbit	1:150
** *Cox41* **	Rabbit	1:250	Goat anti rabbit	1:250
** *BDNF* **	Mouse	1:50	Goat anti mouse	1:50
** *Map2* **	Mouse	1:100	Goat anti mouse	1:100
** *H2A* **	Rabbit	1:100	Goat anti rabbit	1:100
** *S100* **	Rabbit	1:100	Goat anti rabbit	1:100
** *GFAP* **	Rabbit	1:1000	Goat anti rabbit	1:10,000
** *Tub2* **	Mouse	1:100	Goat anti mouse	1:100

## Data Availability

Data are contained within the article.

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
