# Peer review of "Do Microplastics Have Neurological Implications in Relation to Schizophrenia Zebrafish Models? A Brain Immunohistochemistry, Neurotoxicity Assessment, and Oxidative Stress Analysis"

_ijms, 2024, doi:10.3390/ijms25158331_

Round 1

Reviewer 1 Report

Comments and Suggestions for Authors    

The authors investigate the effects of microplastics (MP) using three zebrafish models of schizophrenia. They found that lesions in the optic tectum were more severe and frequent when MP were combined with ketamine or methionine. MP and methionine significantly stimulated the multiplication of progenitor stem cells or radial glial cells in the periventricular region, optic tectum, and cerebellar valve. Methionine exposure led to structural changes, including necrosis in the periventricular zone, across all exposed groups, both alone and in combination. Despite these detrimental effects, methionine also activated periventricular neurogenesis, indicated by the presence of PCNA-positive cells. Additionally, MP exhibited protective effects against oxidative stress, as their combined exposure with methionine and methionine+ketamine led to a successful second antioxidant defense and reduced lipid peroxidation. Overall, the findings suggest complex interactions between microplastics, ketamine, and methionine in neurogenesis and optic tectum lesions.

This work could be publishable in IJMS after the following minor revisions:

  1. Quantification of the immunohistology assays to better understand differences among the treatment groups.

  2. Inclusion of behavioral results to enhance the manuscript.

Author Response

This work could be publishable in IJMS after the following minor revisions:

The study is extremely complex, and we greatly appreciate your consideration. Below you will find the specific answers we have given to the suggestions you made.

Quantification of the immunohistology assays to better understand differences among the treatment groups.

Thank you for suggesting this, we have added a few tables showing the specific differences between the groups. We hope it is now easier to understand.

Inclusion of behavioral results to enhance the manuscript.

This data is part of a larger study being conducted by our research team, and some of the behavioral results have been already published in https://doi.org/10.1016/j.bbr.2023.114742 .

Reviewer 2 Report

Comments and Suggestions for Authors

The aim of the work is unclear in the abstract

Methods, more details about how the enzyme will be determined.

Discussion:

Line 211: n three Schizophrenia models; please explain ?

Discussion is good

 Statistical study is unclearly used

Immunological study was not clearly discussed

The figures are related to the results

Conclusion is representative

References should be updated

Author Response

Thank you for your appreciation, the study is extremely complex. Please find below the specific responses to what you have suggested.

The aim of the work is unclear in the abstract

We rewrote the aim in the abstract. Thank you for this suggestion. 

Methods, more details about how the enzyme will be determined.

We are very grateful for your suggestion. We did a complete rewrite of the paragraph in the manuscript.

Discussion:

Line 211: n three Schizophrenia models; please explain ?

Methionine combined with ketamine is a reliable model of schizophrenia developed by our research team to mimic both positive and negative symptoms, as we previously described in https://doi.org/10.1016/j.bbr.2023.114742.

Discussion is good

Statistical study is unclearly used

Thank you for this suggestion. We completed the paragraph.

Immunological study was not clearly discussed

For a better understanding of the complex immunohistochemical results we have added a few new tables showing the differences between the groups, regarding the discussion the literature is still a bit sacred regarding these aspects. Even more, we completed some aspects in this regard.

The figures are related to the results

Conclusion is representative

References should be updated

We have managed to update the bibliography, although the specialized literature in this field is still scarce and the need for further studies is still very considerable.

Round 2

Reviewer 2 Report

Comments and Suggestions for Authors

The authors succeeded in answer all the reviewer comments